# Wound-inducible ANAC071 and ANAC096 transcription factors promote cambial cell formation in incised Arabidopsis flowering stems

Keita Matsuoka[1], Ryosuke Sato[1], Yuki Matsukura[1], Yoshiki Kawajiri[1], Hiromi Iino[1], Naoyuki Nozawa[1], Kyomi Shibata[1], Yuki Kondo[2,3], Shinobu Satoh[4] & Masashi Asahina [1,5 ✉]

ANAC071 and its homolog ANAC096 are plant-specific transcription factors required for the initiation of cell division during wound healing in incised Arabidopsis flowering stems and Arabidopsis hypocotyl grafts; however, the mechanism remains mostly unknown. In this study, we showed that wound-induced cambium formation involved cell proliferation and the promoter activity of *TDR/PXY* (cambium-related gene) in the incised stem. Prior to the wound-induced cambium formation, both *ANAC071* and *ANAC096* were expressed at these sites. *anac*-multiple mutants significantly decreased wound-induced cambium formation in the incised stems and suppressed the conversion from mesophyll cells to cambial cells in an ectopic vascular cell induction culture system (VISUAL). Our results suggest that ANAC071 and ANAC096 are redundantly involved in the process of "cambialization", the conversion from differentiated cells to cambial cells, and these cambium-like cells proliferate and provide cells in wound tissue during the tissue-reunion process.

[1] Department of Biosciences, Teikyo University, 1-1 Toyosatodai, Utsunomiya, Tochigi, Japan. [2] Department of Biological Sciences, Graduate School of Science, The University of Tokyo, 7-3-1 Hongo, Bunkyo-ku, Tokyo, Japan. [3] Department of Biology, Graduate School of Science, Kobe University, 1-1, Rokkodai-cho, Nada-ku, Kobe, Hyogo, Japan. [4] Graduate School of Life and Environmental Sciences, University of Tsukuba, 1-1-1 Tennodai, Tsukuba, Ibaraki, Japan. [5] Advanced Instrumental Analysis Center, Teikyo University, 1-1 Toyosatodai, Utsunomiya, Tochigi, Japan. ✉email: asahina@nasu.bio.teikyo-u.ac.jp

Plants adapt to the environmental stress by morphological and metabolic change because they cannot move their whole bodies in response to environmental stimuli, such as wounding by herbivory, wind, and man-made cutting. Plants have the ability to heal mechanical wounds. From ancient times, humans have utilized this ability to enable grafting for agriculture and horticulture. However, the molecular mechanism of tissue reunion has not yet been clearly explained.

Wound-induced cambium formation appeared to be extremely important for cell proliferation and the redifferentiation of vascular tissue during the tissue-reunion process, especially in the stem or hypocotyl, because the disconnection of vascular tissue prevents the translocation of photosynthesis products and water, as well as the formation of aerial organs. Auxin promotes cambium formation and differentiation into xylem and phloem elements around the wound site in tobacco stem and cactus graft[1,2]. In incised Arabidopsis flowering stems, vascular cambium regeneration and new vasculature formation were observed following a local auxin response[3]. Many studies have shown that auxin controls the regeneration of xylem and phloem vessels from proliferated cells to repair the function of the vascular system[4–8]. Smetana et al.[9] show that the local accumulated auxin regulates xylem identity and stem-cell organizer function in vascular cambium via the expression of HD-ZIP III transcription factors. Thus, auxin appears to trigger wound-induced cambium formation and the redifferentiation of xylem/phloem vessel elements during the process of wound repair.

The fundamental molecular mechanism of wound-induced cambium formation and redifferentiation during wound healing would be nearly the same as that of cambial proliferation and differentiation during the process of growth and development because genes expressed in the cambium, xylem, and phloem were highly induced during the process of hypocotyl grafting[10]. In developmental growth, the proliferative activity of cambial cells is maintained by CLAVATA3/EMBRYO SURROUNDING REGION-related (CLE) peptide signaling, which has been identified as tracheary element differentiation inhibitory factor (TDIF)[11]. Phloem-derived CLE41 and CLE44 combine with TDIF receptor (TDR, also known as PHLOEM INTERACTED WITH XYLEM (PXY)), which localizes in the cellular membrane of cambial cells, and then promotes its own cell proliferation[12,13]. WUSCHEL-related HOMEOBOX 4 (WOX4) and WOX14 transcription factors act downstream of the auxin–TDR/PXY signaling pathway to promote cell proliferation in the cambium[14,15]. Then, divided cambial cells differentiate to xylem at the inside and phloem at the outside[9]. Arabidopsis NAC DOMAIN CONTAINING PROTEIN (ANAC) encodes plant-specific NAM, ATAF, and CUC (NAC) transcription factor family proteins that are involved in a range of plant developmental processes, including the differentiation of xylem and phloem. VASCULAR-RELATED NAC-DOMAIN 6 (VND6) and VND7, members of the NAC family of transcription factors, are key genes for the differentiation of xylem vessels through the direct regulation of XYLEM CYSTEINE PEPTIDASE 1 (XCP1), which induces programmed cell death for the formation of xylem tracheary elements[16,17]. The BRI1-EMS-SUPPRESSOR 1 (BES1) transcription factor acts upstream of the VND genes; thus, brassinosteroid signaling is also involved in xylem differentiation[18,19]. To maintain cambial cells, BES1 is suppressed by glycogen synthase kinase 3 proteins (GSK3s) downstream of the TDIF–TDR signaling pathway[20]. On the other hand, ANAC086 and ANAC045 were identified as target genes of ALTERED PHLOEM DEVELOPMENT transcription factor, which is required for phloem development and participates in enucleation through exonuclease-domain proteins[21,22]. ANAC020 was identified as an early regulator of phloem

differentiation[23]. Thus, the most ANAC genes are generally known to function in morphogenesis.

A previous study reported that the cell division of pith tissue was initiated by partial incision treatment from 3 days after incision (DAI) in Arabidopsis flowering stems, and ANAC071, which encodes an NAC transcription factor, promotes cell proliferation downstream of an auxin signaling pathway[24]. We have reported that ANAC071 and ANAC096, its close homolog, were induced by cotyledon-derived auxin and redundantly promoted the cell proliferation of vascular tissue during Arabidopsis hypocotyl grafts[25]. However, the mechanisms of ANAC071 and ANAC096 in these processes still remain unclear. To reveal the function of ANAC071 and ANAC096 for the regeneration of wounded tissue, we investigated tissue localization and morphological change in the incised flowering stem, and we concluded that ANAC071 and ANAC096 are essential for wound-induced cambium formation from dedifferentiated cells before the initiation of cell division during the tissue-reunion process.

## Results

**Morphological change in tissue after incision**. Our previous study reported that cell proliferation was observed in pith cells around the incision in partially incised Arabidopsis flowering stems[24], but the cell type and the shape of the divided cells were still unclear. To show morphological and histological changes at the cellular level, we reconstructed a 3D image of the upper region of the incision from serial section at 4 DAI, when the initiation of cell division was observed (Fig. 1a, b). Asymmetric cell division and cell expansion occurred in parenchyma cells of the pith and protoxylem near the incision (Fig. 1c, d). Because the expanded cells caused drastic changes in cellular alignment (Fig. 1a), cross-section observation was also required for the examination of cell proliferation and identification of cell type.

The upper region of the incision visibly expanded outward from 2 to 5 DAI (Supplementary Movie 1 and Fig. 1e, h). At 3 DAI, cortex cells began expansion toward the incised side of the upper region, and endodermal cells showed expansion and cell division (Fig. 1f, g). Moreover, vascular cambial cells showed periclinal cell division and produced multiple cell layers. These cells formed a secondary cell wall, which showed UV-excited autofluorescence and a reticulate pattern at 7 DAI (Fig. 1j and Supplementary Fig. 1a, b); that is, the wound-induced redifferentiation of secondary xylem from proliferated cells in the upper region of the incision.

Using pTDR/PXY::GUS as a cambium marker, we detected the formation of cambial cells in partially incised stems (Fig. 1k–n and Supplementary Fig. 2). The promoter activities of TDR/PXY were detected not only in divided cells in the internal region of vascular tissue at 7 DAI (Fig. 1k, l) but also in periclinal dividing cells in the swelling upper region at 3 DAI (Fig. 1m, n), suggesting that these tissues are composed of de novo cambium-like cells. These TDR/PXY-expressing cambium-like cells were found both in intravascular region and extravascular region.

**Phylogenetic analysis of homologous genes of ANAC071 and ANAC096**. ANAC071 and ANAC096 acted redundantly in tissue reunion in hypocotyl grafts, but the anac071 anac096 double mutant did not completely suppress the cell proliferation of the graft union[25]. Hence, we suspected functional overlap with other ANAC genes. To investigate the functional and evolutionary relationships between ANAC071, ANAC096, and other ANAC genes, phylogenetic analysis was performed using the amino acid sequences of ANAC071, ANAC096, and other NAC genes in Arabidopsis and various plant species. ANAC071, ANAC096, and ANAC011 were located in clades B-III and different

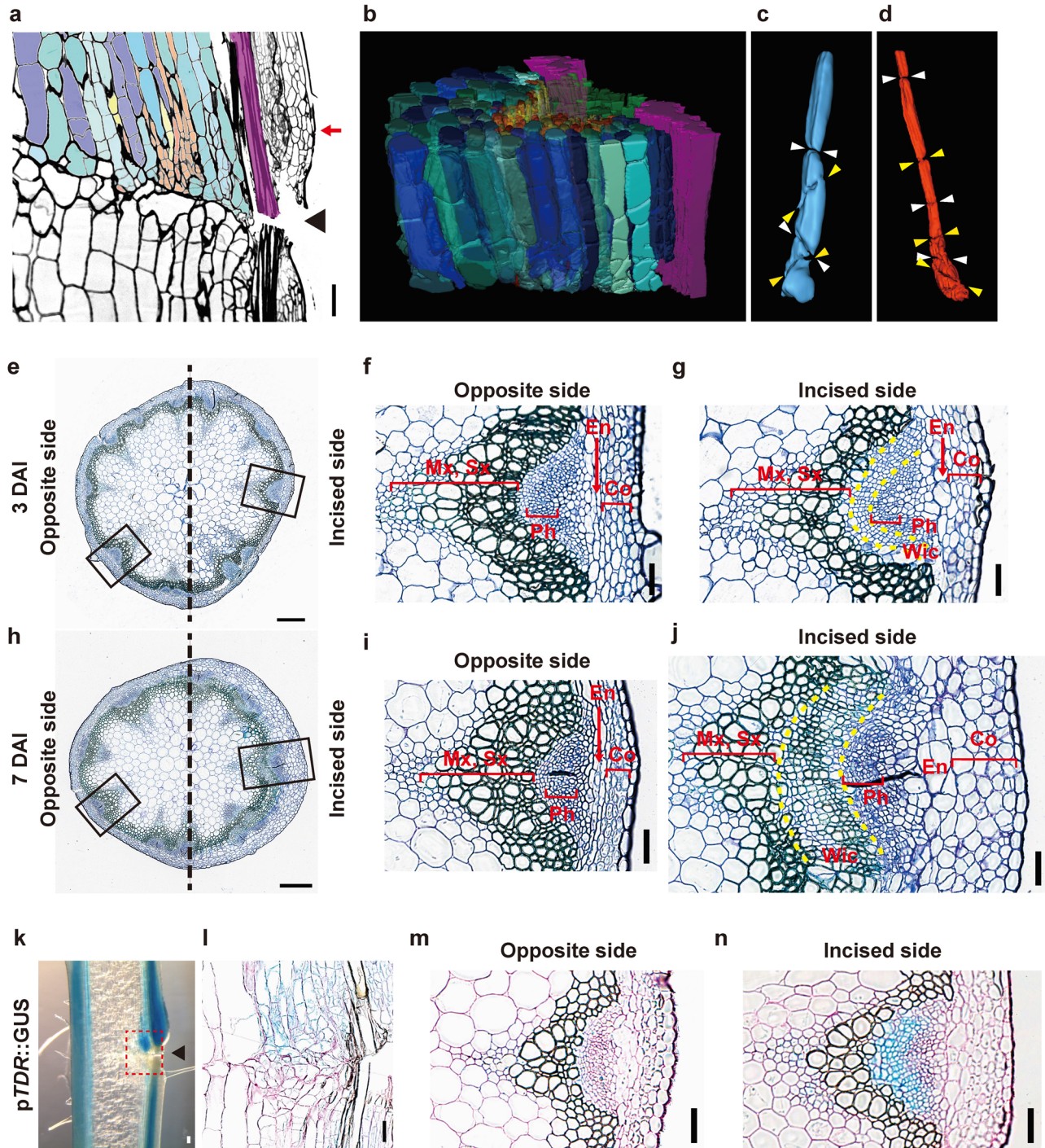

**Fig. 1 Histological observation of incised flowering stem.** (**a–d**) Longitudinal serial sections of WT flowering stem at 4 DAI. Cells are colored by tissue as follows: pith (blue tones), parenchyma cell of protoxylem (yellowish-orange tones), secondary xylem (green), and interfascicular fiber (purple). **a** One of the serial sections. The arrowhead indicates the incision. **b–d** 3D model reconstructed from the upper region of the incision and extracted to strands of pith (**c**) and parenchyma in protoxylem (**d**) derived from one cell. The pair of arrowheads indicate the position of cell division. Thin cross-sections were located in the upper region of the incision (position of red arrow in **a**) at 3 (**e–g**) and 7 DAI (**h–j**). **e**, **h** Opposite and incised sides are shown on the left and right, respectively. Scale bars = 500 μm. The magnified images of the black rectangular area are shown in **f**, **g**, **i**, and **j**. The tissue between two yellow-dashed lines indicates Wic wound-induced cambium. Mx metaxylem, Sx secondary xylem, Ph phloem, En endodermis, Co cortex. **k–n** Histochemical analysis of p*TDR*::GUS in incised flowering stem. **k** Freehand longisection was prepared from stem at 7 DAI. **l** Thin longisection corresponded to the red rectangular area shown in **k**. **m**, **n** Thin cross-sections were prepared from the upper region of the incision at 3 DAI. Cell walls with red color were counterstained by periodic acid-Schiff. Scale bars = 100 μm.

from clade A involved in ANAC045, ANAC086, and ANAC020, which are phloem-related genes (Supplementary Fig. 3), suggesting that ANAC011 has the same function as ANAC071. The relationship between ANAC071 and ANAC096 was closer than

the relationships with NAC genes of other plant species. Clade B-III is mainly composed of NAC genes of eudicots, and all eudicots have NAC genes of clade B-III. In contrast, there are no special rules within clade A.

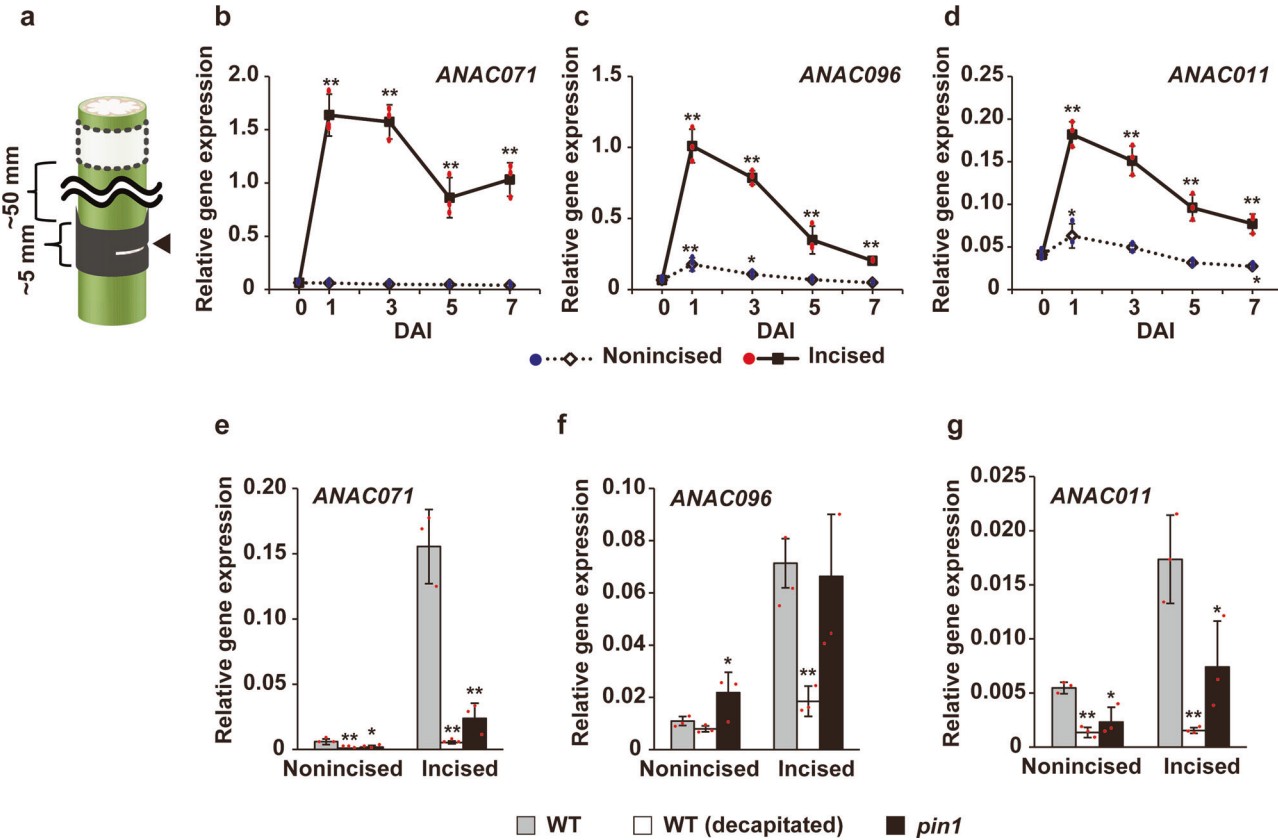

**Fig. 2 ANAC gene expression at 1 to 7 days after incision. a** Schematic model of the sampling region from the incised stem for qRT-PCR analysis of **b–d**. Stem samples (~5 mm length) were separately harvested from the region including the incision (black) and from the nonincised region located ~50 mm above the incision (white). Arrowheads indicate the incision. **b–d** Relative gene expression of *ANAC071*, *ANAC096*, and *ANAC011* was analyzed in flowering stems at 1–7 days after incision (DAI). **e–g** The relative expression levels of *ANAC071*, *ANAC096*, and *ANAC011* were analyzed in flowering stems at 3 DAI. *pin1* mutants were confirmed by T-DNA insertion with PCR and by the phenotype of the pin-formed flowering stem. Values are normalized by *ACT2* expression and represent the means of triplicate experiments ± SD. Sampling time at 0 DAI means stem before incision (harvested from same height as incised region; black box in **a**). Asterisks indicate statistically significant differences compared with 0 DAI or WT (*$P < 0.05$, **$P < 0.01$; Dunnett's test).

**Gene expression analysis of ANAC in incised flowering stems.** We examined the gene expression of *ANAC071*, *ANAC096*, and *ANAC011* using quantitative reverse transcription PCR (qRT-PCR) analysis in incised flowering stems (Fig. 2a). *ANAC071* expression was drastically upregulated in the incised region from 1 to 7 DAI (~25-fold increase from intact stem) (Fig. 2b). The gene expression levels of *ANAC096* and *ANAC011* were increased ~15 and ~4-fold, respectively, at the incised region at 1 DAI and then gradually decreased until 7 DAI (Fig. 2c, d). The expression of *ANAC011* was induced after incision, but its transcript level was only approximately one-tenth of that of *ANAC071* or *ANAC096*. The upregulation of *ANAC071*, *ANAC096*, and *ANAC011* was suppressed by decapitation at the incised region at 3 DAI (Figs. 2e–g and 3b, c, h, i). A deficient mutant of PIN1, a polar auxin transporter, showed suppression of *ANAC071* and *ANAC011* but no significant difference in *ANAC096* (Fig. 2e–g).

The localizations of *ANAC071* and *ANAC096* were histochemically examined by promoter::GUS analysis in incised flowering stems (Fig. 3 and Supplementary Fig. 2). The promoter activity of *ANAC071* was mainly observed in the upper region of the incision, while that of *ANAC096* was detected around the incision at 3 DAI (Fig. 3a, b, g, h and Supplementary Fig. 2b, c). In cross-sectional observation of the upper region of the incision, *ANAC071* and *ANAC096* were expressed only on the incised side (Fig. 3d, j). *ANAC071* was extensively expressed in tissues including parenchyma cells (of the pith, protoxylem, metaxylem, and secondary xylem), phloem, endodermis, and cortex on the

incised side (Fig. 3e). Moreover, GUS staining of *ANAC071* was also shown in the wound-induced cambium of intravascular region. The promoter activity of *ANAC096* was detected in wound-induced cambium of intravascular region, phloem, and endodermis at 3 DAI (Fig. 3k). At the cross-section of the incised position, *ANAC071* and *ANAC096* were commonly expressed in parenchyma cells of the pith and protoxylem (Fig. 3f and l).

We also examined the expression of *ANAC* genes among organs in seedlings. *ANAC071* expression was scarce in all organs of the intact plant but greatly enhanced by cuts in the hypocotyl (Supplementary Fig. 4a, d). *ANAC096* showed high expression at the shoot meristem and cut hypocotyl (Supplementary Fig. 4b, d). The basal expression level of *ANAC011* was lower than that of *ANAC071* and *ANAC096*, but its expression was induced in the cut hypocotyl (Supplementary Fig. 4c).

**Morphological analysis of incised flowering stem in anac mutants.** To reveal the physiological function of *ANAC* genes, the histological morphologies of incised flowering stems of WT and *anac* mutants were analyzed. *anac071 anac096 anac011* multiple mutants showed no visible phenotype in 6-day-old and 5-week-old seedlings (Supplementary Fig. 5a, b). In WT plants, cell division was observed in the parenchyma cells of the pith and protoxylem near the incision at 7 DAI (Supplementary Fig. 6a). Cell proliferation was also observed around the incisions of *anac071* and *anac096* single mutants, while cell proliferation was suppressed near the incisions of *anac071 096* double and *anac071*

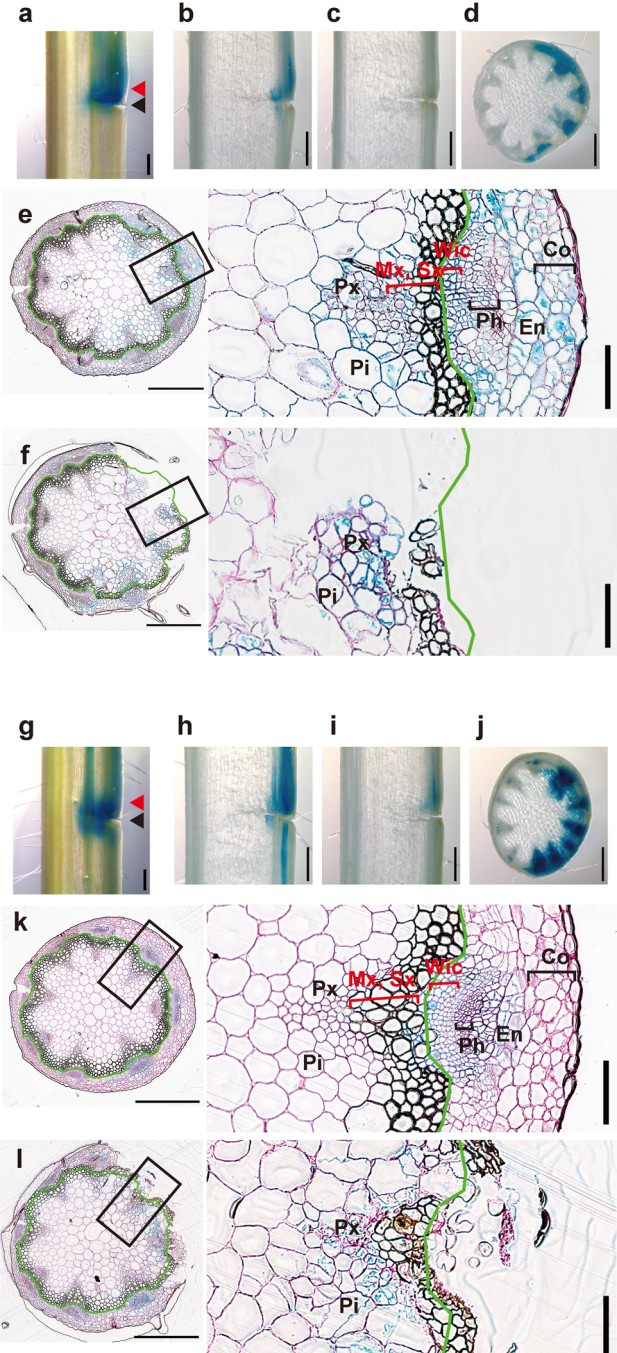

**Fig. 3 Tissue localization of *ANAC* gene expression in incised stems.**
Histochemical analysis of pANAC071::GUS (**a–f**) and pANAC096::GUS (**g–l**) was performed at 3 DAI. Opposite and incised sides are shown on the left and right, respectively. **a**, **g** Incised stems are displayed from the surface. Freehand longisections were prepared from incised stems with (**b**, **h**) or without (**c**, **i**) apical buds. **d**, **j** Freehand cross-sections from the incision to the upper part. Thin cross-sections were prepared from the upper region of the incision (**e**, **k**; red arrowheads of **a** and **g**) and the incised region (**f** and **l**; black arrowheads of **a** and **g**). Scale bars = 500 μm. Magnified images of the black rectangular area are shown on the right. The green line shows the outline of sclerenchyma tissue. Pi pith, Px protoxylem, Mx metaxylem, Sx secondary xylem, Wic wound-induced cambium, Ph phloem, En endodermis, Co cortex. Cell walls with red color were counterstained by periodic acid-Schiff. Scale bars = 100 μm.

*096 011* triple mutants, resembling the morphology of decapitated WT plants (Supplementary Fig. 6a–f). A previous study reported that decapitation (removal of the apical bud) strongly suppressed cell division during the tissue-reunion process due to inhibition of the auxin supply to the incised site[24]. From the observation of the cross-section at the incised position, the size of the cells on the incised side in the decapitated WT, *anac071 096* double mutant, and *anac071 096 011* triple mutant were larger than those in the WT, *anac071* mutant, and *anac096* mutant (Supplementary Fig. 6g–l). In this regard, there was no significant difference between the *anac071 096* double mutant and the *anac071 096 011* triple mutant.

As described above, both *ANAC071* and *ANAC096* were induced in these regions after incision. Hence, the cell layers of these wound-induced cambial cells in the upper region of the incision were also quantitatively compared (Fig. 4), and DAPI-staining and EdU-labeling experiments were also performed to confirm the cell proliferation (Supplementary Fig. 7). In the WT plants, these wound-induced cambium consisted of ~9 layers of cells at 7 DAI, and the number of layers was reduced to ~5 by decapitation (Fig. 4a, b). The number of cell layers in *anac071* was similar to that in WT, but *anac096* showed significantly fewer cell layers than the WT. Moreover, the numbers of cell layers in *anac071 096* double and *anac071 096 011* triple mutants were decreased to almost the same degree as in the decapitated WT, although no differences in secondary xylem formation on the non-incised side or in vascular development under ordinary growth conditions were observed in the *anac* mutants (Fig. 4a and Supplementary Fig. 8). In the incised region of WT at 5 DAI, higher number of DAPI-stained nuclei were observed, and these cells were strongly labeled with EdU (Supplementary Fig. 7). On the other hand, relatively small numbers of DAPI-stained nuclei with EdU-labeling were detected in incised stem of *anac071 096 011* triple mutants and non-incised region of WT (Supplementary Fig. 7). More than 80% of cell layers had a secondary cell wall in WT (Fig. 4c). These ratios in all *anac* mutants were significantly lower than in WT but higher than in decapitated WT.

**Ectopic vascular cell induction assay using cotyledon.** Given that wound-induced xylem formation was partially suppressed in the incised flowering stem of *anac* mutants, we examined the function of *ANAC* genes in the regulation of cell differentiation into xylem using a culture system named "vascular cell induction culture system using Arabidopsis leaves" (VISUAL)[23]. Defects in xylem differentiation-related genes, such as *BES1* and *VND6*, caused a decrease in ectopic xylem formation under the VISUAL assay[18]. Ectopic xylems were observed in both *anac071* and *anac011* single mutants, similar to the WT (Fig. 5b and Supplementary Fig. 9a). Ectopic xylem elements were significantly decreased in the *anac096* single mutant and markedly in the *anac071 096* double and *anac071 096 011* triple mutants (Fig. 5a, b and Supplementary Fig. 9a). To observe the shape of cells, optical sections were obtained from pseudo-Schiff propidium iodide–stained cotyledons using confocal laser-scanning microscopy. In WT, a large number of tracheary elements were formed at mesophyll cells by the addition of bikinin (Supplementary Fig. 9b). Phloem sieve element-like cells, which undergo multiple cell divisions, as reported in a previous study[23], were also observed among mesophyll cells. In contrast, tracheary elements were localized near veins in the *anac071 096 011* triple mutant.

*ANAC071* expression was synergistically increased after 2 days of culture (DOC) by the simultaneous addition of 2,4-dichlorophenoxyacetic acid (2,4-D) and kinetin and/or bikinin,

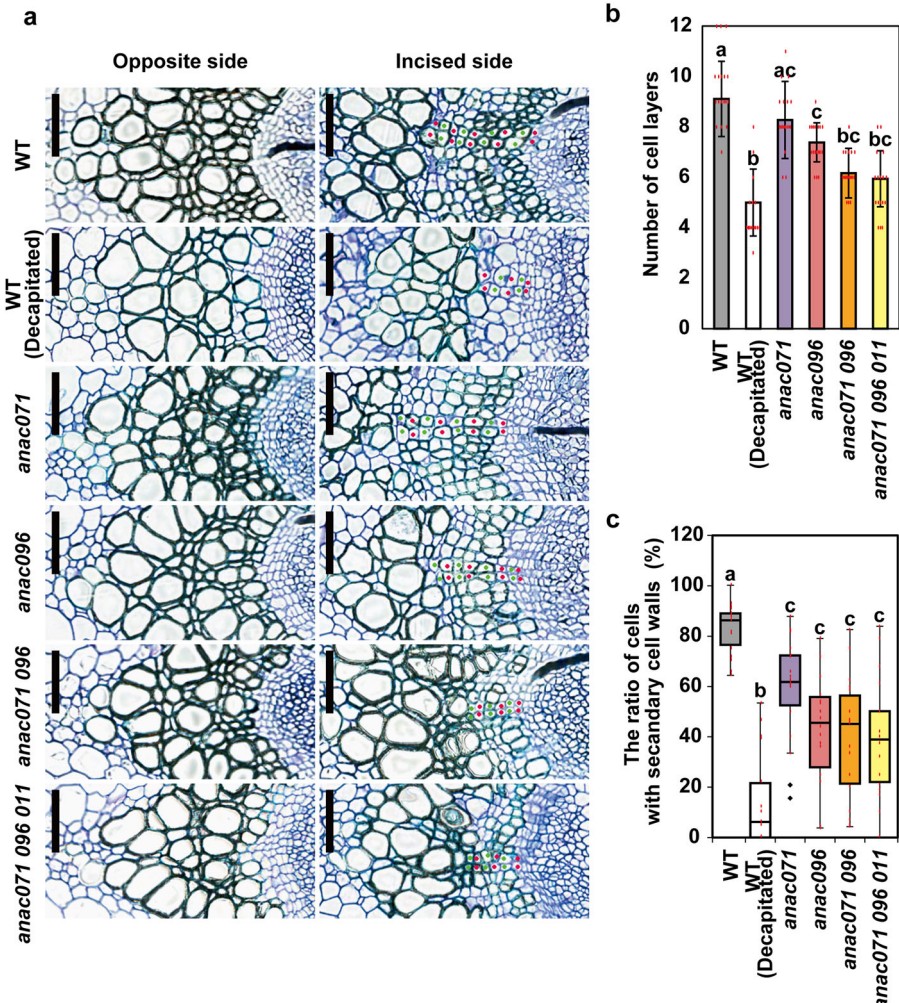

**Fig. 4 Histological observation of the secondary xylem in incised stems of *anac* mutants. a** Thin cross-sections of vascular tissue at the upper end of the incision at 7 DAI. Left, opposite side; right, incised side. Layers of periclinally divided cells were counted as red and green dots on the incised side (bottom). Scale bars = 100 μm. **b** Bars are the mean number of cell layers at 7 DAI (*n* = 18 from 9 plants). Different letters indicate statistically significant differences (*P* < 0.05; Tukey's test). **c** Boxplots showing the ratio of cells with secondary cell walls in the cell layer. Statistical analysis was performed using the Steel–Dwass test (*P* < 0.05).

while *ANAC096* expression was not induced by 2,4-D alone but was induced by its simultaneous application with kinetin and bikinin (Fig. 6a, b). However, *ANAC011* was not induced under any culture conditions and was expressed at a low basal level (Fig. 6c). The promoter activities of *ANAC071* and *ANAC096* were detected in the entire cotyledon at 4 DOC by bikinin application (Fig. 6d). These inductions of *ANAC071* and *ANAC096* occurred even in the *bes1-1* mutant (Supplementary Fig. 10a, b).

Next, the expression levels of tissue-specific genes were compared between the WT and the *anac071 096 011* triple mutant. *XCP1* and *VND6*, xylem-related genes, were upregulated in bikinin-treated cotyledons of the *anac071 096 011* triple mutant, similar to the WT; however, *TDR/PXY* and *SIEVE ELEMENT OCCLUSION-RELATED 1* (*SEOR1*), phloem-related genes[26] were significantly suppressed in the *anac071 096 011* triple mutant (Fig. 6e, f and Supplementary Fig. 10e, f). *MONOPTEROS* (*MP*), an auxin response factor and a cambium-related gene, was not significantly different in the *anac071 096 011* triple mutant (Supplementary Fig. 10d). The GUS activity of the promoter *TDR/PXY* was localized near the leaf veins in WT plants before induction, and then it was detected

in the whole cotyledon at 4 DOC upon the addition of bikinin (Fig. 6g), similar to the results from previous work using leaf disc culture[23]. The tissue localization of GUS in the *anac071 096 011* triple mutant showed the same pattern as in the WT before induction or culture without bikinin after 4 DOC, and GUS activity was detected only in the leaf vein in the presence of bikinin (Fig. 6g). The decomposition of chlorophyll and suppression of rubisco small subunit (*RBCS1B*) occurred both in the WT and *anac071 096 011* triple mutant after 2 DOC with bikinin (Supplementary Fig. 10g, h).

**Expression analysis of tissue-specific gene in incised flowering stem.** Based on the results of the VISUAL assay, we focused on the changes in the expression of vascular formation-related genes in the incised flowering stem. The expression of cambium-related genes (*TDR/PXY*, *WOX4*, and *MP*), xylem-related genes (*VND7* and *XCP1*), and phloem-related genes (*SEOR1*) was induced in the incised region at 1 DAI and then increased until 3 DAI (Supplementary Fig. 11a–f). These tissue-specific genes, auxin response gene (*IAA5*), cell division marker gene (*CYCB1;1*), and *CLE* genes were significantly decreased in the incised region by decapitation treatment for the depletion of auxin (Supplementary

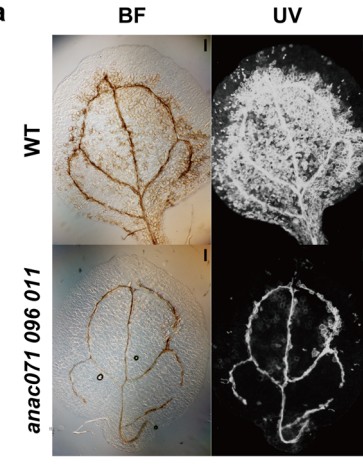

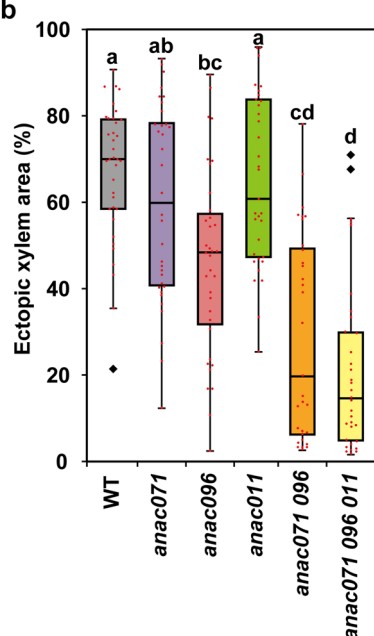

**Fig. 5 Morphology of cotyledons under VISUAL assay. a** Observation of ectopic xylem vessel elements in WT and *anac071 096 011* triple mutant at 4 DOC. Left, bright-field images; right, UV autofluorescence images. Scale bars = 200 µm. **b** Boxplots showing the percentage of ectopic xylem area in WT and *anac* mutants (n = 36 for WT, 34 for *anac071*, 35 for *anac096*, 33 for *anac011*, 31 for *anac071 096*, and 33 for *anac071 096 011*, respectively). Diamond marks indicate outliers. Statistical analysis was performed using the Steel–Dwass test ($P < 0.05$).

Fig. 12a–j). The expression of these genes was also suppressed in the *pin1* mutant, except for *MP* and *IAA5*.

We compared the expression of these genes between WT and *anac* mutants at 3 DAI. Although the gene expression of *XCP1* and *VND7* in the incised region was not affected in *anac* mutants, *TDR/PXY* and *WOX4* expression was significantly suppressed in the *anac071 096* double mutant and *anac071 096 011* triple mutant (Fig. 7a–c and Supplementary Fig. 12f). The promoter activity of *TDR/PXY* was strongly suppressed in the pith and xylem parenchyma around the incision in the *anac071 096 011* triple mutant (Fig. 7e; red arrowheads). In contrast to the WT, the promoter activity of *TDR/PXY* and cell division were not observed in the pith and xylem parenchyma of the *anac071 096 011* triple mutant (Fig. 7f). GUS staining was slightly decreased in the upper region of the incision (Fig. 7e; red asterisks) due to the

decrease in the GUS-stained cell layers (Fig. 7g). *CLE41* and *CLE44*, which are promoting factors of the TDR/PXY-WOX4 signaling pathway, were not suppressed in *anac* mutants but were markedly suppressed in decapitation samples and *pin1* mutants (Supplementary Figs. 12i, j and 13h, i). The upregulation of *CYCB1;1* and *SEOR1* was also suppressed in the *anac071 096* double mutant and the *anac071 096 011* triple mutant (Fig. 7d and Supplementary Fig. 13g). However, *MP* and *IAA5* gene expression was not significantly suppressed in *anac* mutants (Supplementary Fig. 13d, e).

## Discussion
Prior to redifferentiation, cambium-like tissues are newly formed around the wound tissue. Several previous studies reported that the parenchyma cells around the wound site divided and redifferentiated into xylem/phloem vessels to connect the vascular system after incision in eudicots[27–29]. Nishitani et al.[30] reported that cambial activity occurred around wound sites in the incised *Zinnia* internode, and they inferred that unknown vascular-derived factors are required for the formation of cambial cells. Recent studies reported that the spatial pattern of the auxin response gene was similar to that of the cambium marker gene around the wound site in Arabidopsis[3,10], suggesting that wound-induced cambium formation was associated with auxin signaling[1,31,32]. We showed that the parenchyma cells of the pith/protoxylem showed *TDR/PXY* promoter activity and initiated cell division around the incised site (Fig. 1c, d, k, l). The upregulation of *TDR/PXY* and *CYCB1;1* was strongly suppressed in the incised region by decapitation and *pin1* mutation (Supplementary Fig. 12c, e), which suppressed cell proliferation around the wound site[24] (Supplementary Fig. 6d, j). *PIN1* was polarly localized in vascular cambial cells under ordinary conditions but showed new expression in pith cells around the incised site after wounding[3]. Auxin was transported into parenchyma cells around the wound site via newly expressed *PIN1*, and then these cells acquired cambial behavior and proliferated. These combined results indicated that parenchyma cells around the wound site showed high cell proliferation activity with cambial status after incision, and auxin transportation and localization played an important role in this process. We named these phenomena "cambialization", which means the conversion of differentiated cells to cambial cells and /or activation of cambial cell proliferation in the tissue repair process of incised flowering stems.

Previous reports have shown that *ANAC071* was induced by auxin, which accumulated in the upper region of the incision and promoted cell proliferation around the wound site[24,33]. Moreover, *ANAC071* and *ANAC096* were induced by cotyledon-derived auxin and acted redundantly to promote the cell proliferation of vascular tissue in the hypocotyl graft[25]. We found that *ANAC071* and *ANAC096* expression was induced earlier than that of *TDR/PXY* in two tissues where wound-induced cambium formed in the parenchyma cells of the pith/protoxylem and the periclinal dividing cells of the upper region (Figs. 1l, n, 2b, c, 3e, f, k, l, and Supplementary Fig. 11a). The *anac*-multiple mutant showed a significant reduction in cell proliferation and *TDR/PXY*-expressing cells in these tissues. (Figs. 4a, b and 7e–g). These results indicate that these *ANAC* genes are involved in wound-induced cambium formation through auxin signaling during the tissue-reunion process. We hypothesized that ANAC011 acts redundantly with ANAC071 and ANAC096, but there were no differences in morphology and *CYCB1;1* expression in the incised region between the *anac071 096* double mutant and the *anac071 096 011* triple mutant (Figs. 4a–c, 7d, and Supplementary Fig. 6e, f, k, l). ANAC011 is not critical for tissue reunion because the

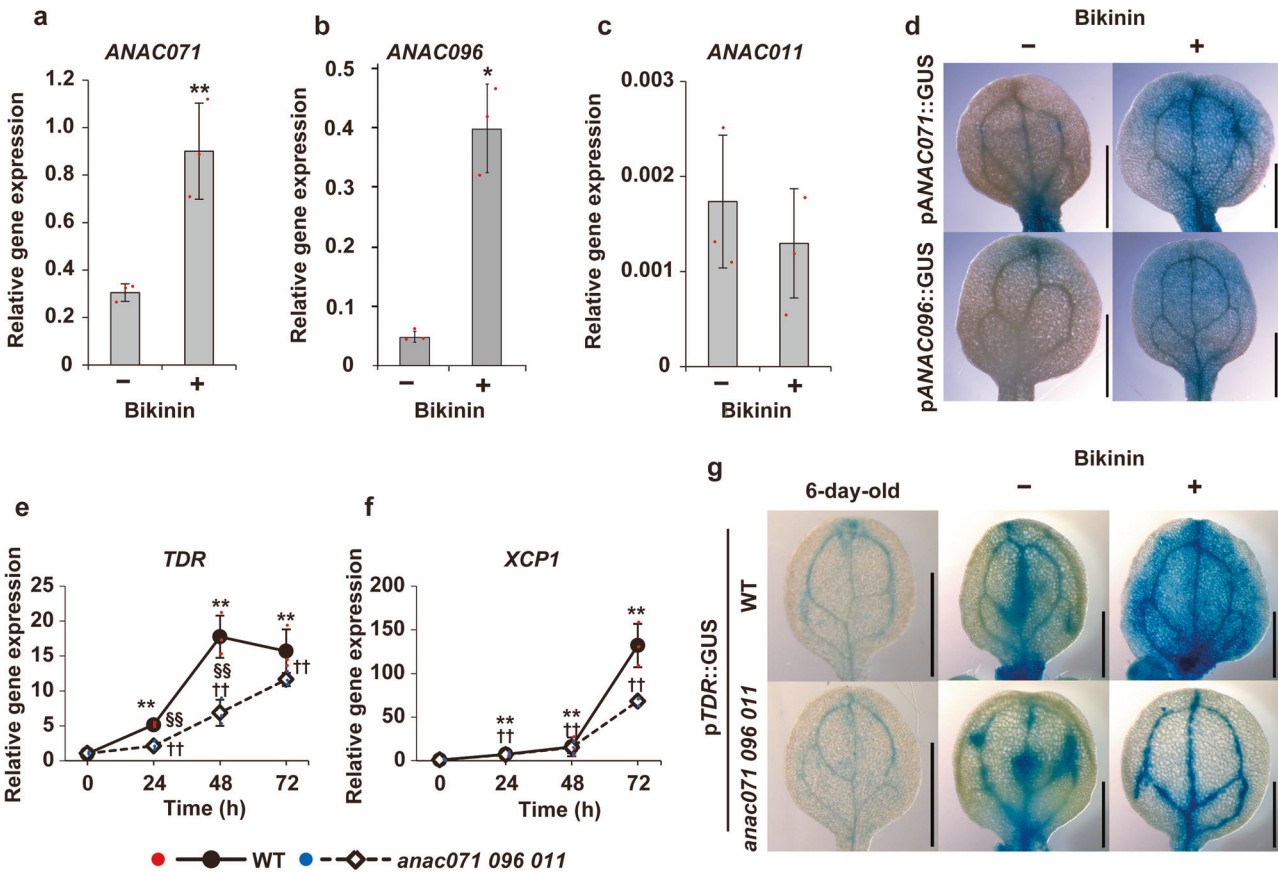

**Fig. 6 Gene expression of *ANAC* and *TDR/PXY* under VISUAL assay. a–c** *ANAC071* (**a**), *ANAC096* (**b**), and *ANAC011* (**c**) gene expression with (+) or without (−) bikinin treatment. Above-ground parts of WT were cultured under normal growth conditions for 2 DOC and used for RNA extraction. Statistical analysis was performed using Welch's test (**P* < 0.05. *** P* < 0.01). **d** Histochemical GUS staining of p*ANAC071*::GUS (top) and p*ANAC096*::GUS (bottom). Cotyledons were observed at 4 DOC. Scale bars = 1 mm. **e, f** *TDR/PXY* and *XCP1* expression was examined in the WT (close circles) and the *anac071 096 011* triple mutant (open diamonds) at 0, 24, 48, and 72 h of culture. Relative gene expression was calculated when WT 0 h was set to 1. Asterisks or daggers indicate statistically significant differences from 0 h in the WT or *anac* mutant, respectively (** (††) *P* < 0.01; Tukey's test). Red section signs indicate significant differences between WT and *anac* mutants. **g** Histochemical GUS staining from p*TDR/PXY*::GUS in WT and *anac071 096 011* triple mutant background. Cotyledons were displayed in 6-day-old plants and at 4 DOC with or without bikinin. Scale bars = 1 mm.

basal expression level of *ANAC011* was substantially lower than that of *ANAC071* and *ANAC096* (Fig. 2b, d).

Another possible explanation is that cell division occurred from original cambial cells by the accumulation of auxin but not through ANAC071 and ANAC096. Periclinal dividing cells were observed in flowering stems after treatment with auxin or its transport inhibitor, and this auxin-dependent division activity was completely suppressed in *wox4* and *tdr* mutants[14,34]. In the incised stem, *WOX4* upregulation was markedly suppressed by decapitation and *PIN1* mutation, while *anac*-multiple mutants showed a slight but significant decrease (Fig. 7c and Supplementary Fig. 12f). *anac*-multiple mutants did not alter the expression of auxin-responsive genes or *CLE41/44* in the incised region (Supplementary Fig. 13d, e, h, i). If ANAC071 and ANAC096 play a role in the cambialization process in xylem parenchyma cells after incision (Fig. 8a), it is no wonder that *anac*-multiple mutants show auxin-dependent mitotic activity in original cambial cells. Xylem parenchyma cells exist as initial developing xylem cells between cambial cells and lignified xylem cells in flowering stems[35,36]. ANAC071 and ANAC096 might promote cambial cell behavior of the initial xylem cells and result in increased cell proliferation (Fig. 8b). The mechanisms of these process are far more efficient than activating a limited number of original cambial cells for high cell proliferation rates after wounding. However, these proposed mechanisms are very

difficult to demonstrate experimentally, such as by live cell imaging deep inside the tissue of a flowering stem.

Overexpression lines of *ANAC071* and *ANAC096* were observed to exhibit almost normal phenotypes under ordinary growth conditions[33,37]. Thus, the upregulation of *ANAC071* and *ANAC096* alone was not enough to convert from dedifferentiated cells into cambial cells, and any factor may be required for cambialization.

We examined the function of ANAC genes in the ectopic vascular cell differentiation process using VISUAL in this study. Ectopic xylem formation was markedly suppressed in *anac*-multiple mutants compared to *anac* single mutants (Fig. 5a, b and Supplementary Fig. 9a, b). There was no difference between the *anac071 096* double mutant and the *anac071 096 011* triple mutant, and *ANAC011* was not upregulated by the induction of ectopic vessels (Figs. 5b and 6c). These results show that the ANAC071 and ANAC096 genes are redundantly involved in ectopic xylem formation. *XCP1* and *VND6* expression was induced even in the *anac071 096 011* triple mutant, while *TDR/PXY* was significantly suppressed (Fig. 6e, f and Supplementary Fig. 10e). In the *bes1* mutant, which has defects in xylem differentiation, the expression of *ANAC071* and *ANAC096* was induced similarly to that in the WT (Supplementary Fig. 10a, b). These results suggested that the suppression of ectopic xylem formation was caused by the defect of ectopic cambial

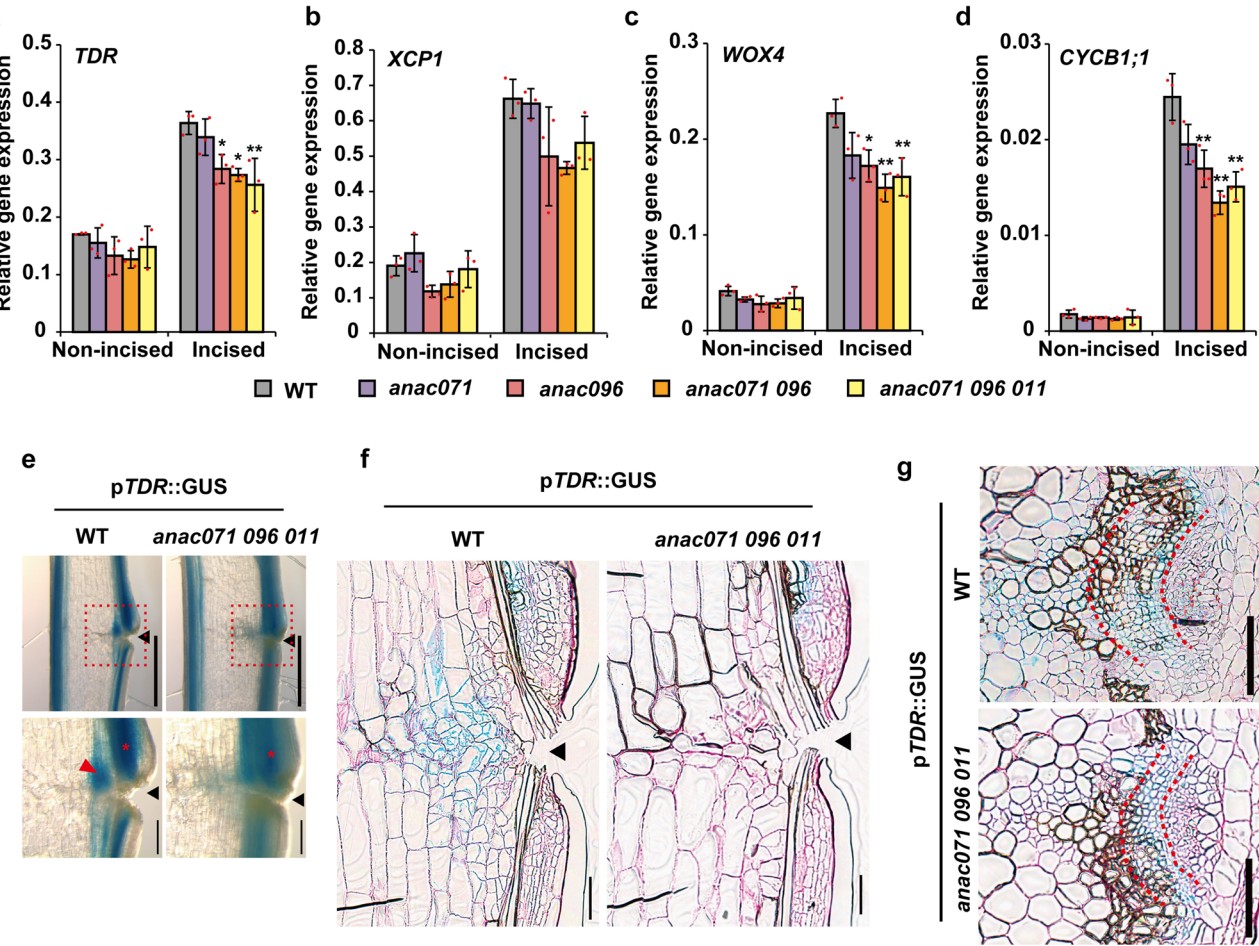

**Fig. 7 Comparison of wound-induced cambium between WT and *anac* mutants. a–d** The relative expression levels of *TDR/PXY*, *XCP1*, *WOX4*, and *CYCB1;1* in flowering stems were analyzed at 3 DAI. The means ± SD of triplicate batch experiments are shown. Statistical analysis was performed using Dunnett's test, in comparison with WT (**p* < 0.05). **e–g** Histochemical p*TDR/PXY*::GUS staining in the incised flowering stems of the WT and the *anac071 096 011* triple mutant background at 7 DAI. **e** Freehand longisections. Scale bars = 1 mm. The regions in red dotted-line boxes are shown at higher magnification (bottom, Scale bars = 200 μm). Red arrowhead, wound-induced cambium of extravascular region; red asterisks, wound-induced cambium of intravascular region; black arrowheads, incised site. **f** Thin longisections located between vascular bundles. **g** Vascular bundles in the upper region of the incision. The area between two red dashed lines indicates wound-induced cambium of intravascular region. Cell walls with red color were counterstained by periodic acid-Schiff. Scale bars = 100 μm.

cells in *anac*-multiple mutants (Fig. 8c). Melnyk et al.[10] proposed that the tissue-reunion process is closely related to the VISUAL assay because the differentially expressed genes were highly overlapped among the transcriptome data of Asahina et al.[24] and Kondo et al.[23]. In the VISUAL assay, mesophyll cells changed their fate to that of cambial cells before the differentiation of xylem cells[18]. This phenomenon, the differentiation of parenchyma cells of the pith/protoxylem in incised flowering stems, is highly similar to the formation of ectopic xylem differentiation in this study(Supplementary Fig. 1). The VISUAL assay results strongly support the hypothesis that ANAC071 and ANAC096 contribute to ectopic vascular cell formation during the tissue-reunion process (Fig. 8a, b).

In conclusion, we performed a detailed analysis of the cellular morphology and localization of gene expression in incised flowering stems and of the function of ANAC genes in ectopic vascular cell differentiation using VISUAL. ANAC071 and ANAC096 act as inducers of cell proliferation through auxin signaling during the tissue-reunion process. Consequently, cell division was initiated in wound-induced cambium, redifferentiated into xylem and phloem vessels, and then the wounded tissue was healed. In this study, we showed that ANAC071 and

ANAC096 were involved in the process of cambialization in incised flowering stem; however, direct evidence to support cell dedifferentiation/ redifferentiation, and the molecular mechanisms underlying the regulation of ectopic differentiation into cambial cells, cell proliferation and vascular differentiation, should be further examined.

## Methods

**Plant materials and growth conditions**. We used *Arabidopsis thaliana* of ecotype Columbia-0 as the WT. The following transgenic lines have been previously described: *anac071* (SALK_012841), *anac096* (SALK_078797), *anac071 096* double mutants, p*ANAC071*::GUS and p*ANAC096*::GUS[25]; *bes1-1* and p*TDR/PXY*::GUS[20]. Seeds of the *anac011* (GABI_752H11) line were purchased from the Nottingham Arabidopsis Stock Center (Nottingham, UK) and crossed with *anac071 096* double mutants to create the *anac071 096 011* triple mutant. Seeds of the *pin1* (SALK_097144) line were purchased from the Arabidopsis Biological Resource Center. The presence of T-DNA insertion was verified by the amplification of PCR products using the primer pair in Supplementary Table 1.

Seeds were surface-sterilized with sodium hypochlorite containing 1% (w/v) active chloride for 6 min and imbibed in darkness at 4 °C overnight. Imbibed seeds were sown on the pots filled with mixed soil (1:3 [v/v] vermiculite [Nittai, Japan]: Genki-kun No. 2 [Katakura & Co-op Agri]) and grown in a Plant Cultivation Rack (Tomy Seiko) under white light (80 μmol m$^{-2}$ s$^{-1}$) at 22 °C. For the incision experiment, a mature flowering stem was incised at ~3 cm from the base to one-half of its diameter using a microsurgical knife (Surgical Specialties) under a

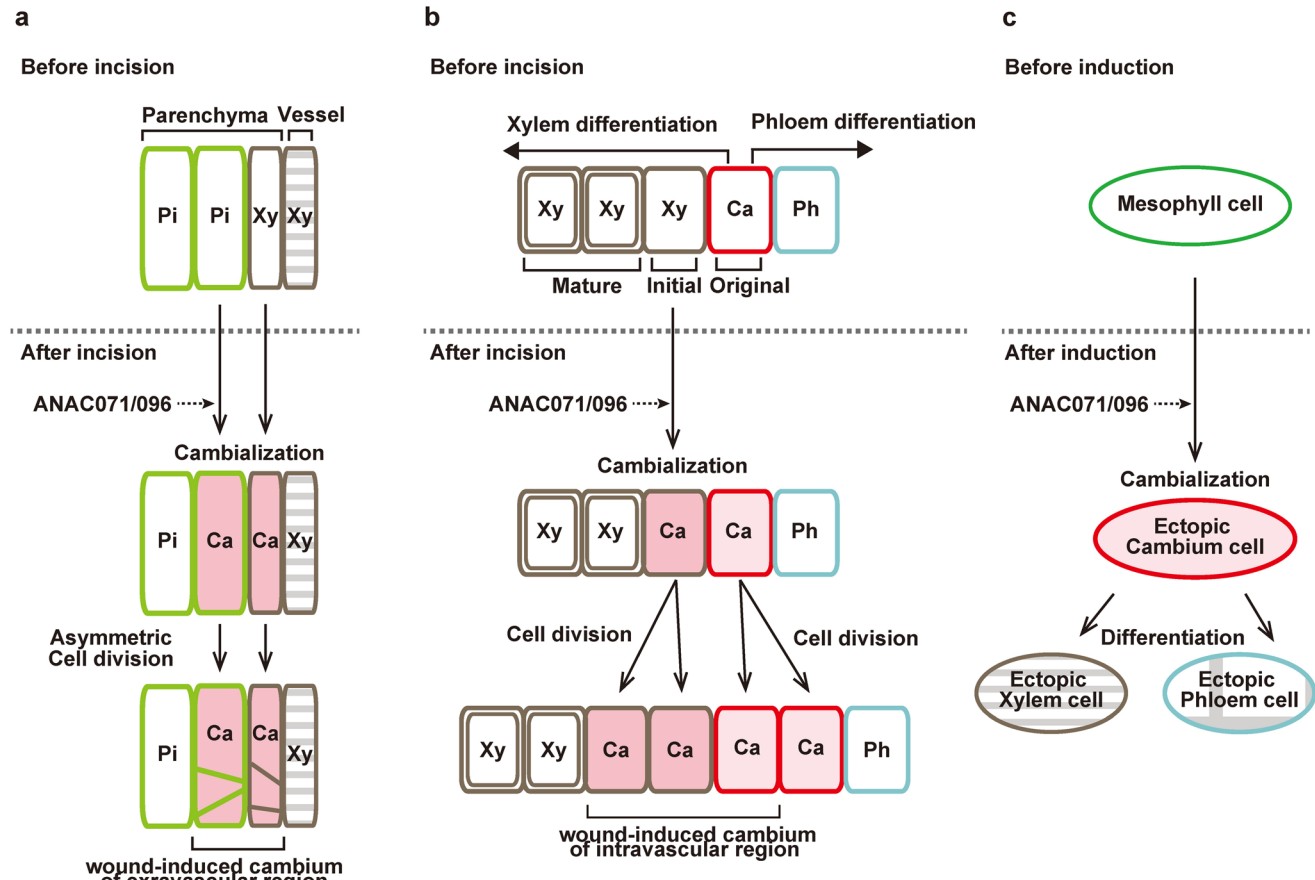

**Fig. 8 Schematic model of cambialization process by ANAC071 and ANAC096. a** Parenchyma cells of pith and protoxylem are converted to cambial cells around the incised site by ANAC071 and ANAC096. These cambium-like cells asymmetrically divide and become to wound-induced cambium of extravascular region. **b** Original vascular cambial cells divide and differentiate into xylem and phloem cells before incision. Initial xylem parenchyma cells are converted to cambial cells around the incised site by ANAC071 and ANAC 096. These cambium-like cells and original cambial cells periclinally divide and become wound-induced cambium of intravascular region. Pi pith, Xy xylem, Ca cambium, Ph phloem. **c** In the VISUAL assay, mesophyll cells are converted to cambial cells and then differentiate into ectopic xylem or phloem cells.

stereomicroscope, according to previous study[24]. Incised plants were grown for up to 7 days under normal growth conditions. To examine the effect of decapitation on tissue reunion, the main stem was cut 10 cm from the base, and the lateral buds and cauline leaves were removed.

**qRT-PCR analysis.** Segments of flowering stems (from ~8 plants) were harvested in liquid nitrogen using a microsurgical knife (Surgical Specialties), as described in Fig. 1a, and then ground to a fine powder with a ceramic mortar and pestle under liquid nitrogen. Total RNA was extracted using QIAshredder (Qiagen) and the RNeasy Plant Mini Kit (Qiagen). cDNA was prepared from 1 μg of total RNA using the PrimeScript RT Reagent Kit with gDNA Eraser (Takara). For VISUAL assay samples, total RNA was extracted with the RNeasy Micro Kit (Qiagen) from ~15 plants, and 100 ng of RNA was used for reverse transcription. qPCR was performed using a 7500 Fast Real-Time PCR system (Thermo Fisher Scientific) with a 20 μl total reaction volume (10 μl of Fast SYBR Green Master Mix (Thermo Fisher Scientific), 1 μl of cDNA and 0.2 μM each primer; Supplementary Table 1). Relative gene expression was normalized to the expression of *ACT2* and calculated from three independent replicates.

**Histochemical analysis of GUS activity.** Plant samples were submerged in ice-cold 90% (v/v) acetone for 5 min. After washing with distilled water, samples were vacuum infiltrated with staining solution (100 mM sodium phosphate buffer pH 7.0, 10 mM EDTA, 0.1% [v/v] Triton X-100, 0.5 mM $K_3Fe(CN)_6$, 0.5 mM $K_4Fe$ $(CN)_6$, 2 mM 5-bromo-4-chloro-3-indolyl-ß-glucuronide, and 20% [v/v] methanol), and then the infiltrated samples were incubated at 37 °C for 1 h. Stained samples were submerged in 70% (v/v) ethanol and observed using a stereomicroscope (M165FC, Leica) with a MC120 HD camera (Leica).

**Light microscopic observation of section.** Samples of incised flowering stem were fixed in mixtures of 4% (w/v) paraformaldehyde and 2.5% (v/v) glutaraldehyde in 0.1 M phosphate buffer (pH 7.4) overnight at 4 °C. After washing with phosphate

buffer, samples were dehydrated through graded ethanol solutions of 30, 50, 90, and 100% twice (v/v) for 30 min each. For GUS-stained samples, dehydration was started with 70% (v/v) ethanol. Dehydrated samples were embedded in Technovit 7100 resin (Heraeus Kulzer GmbH) and sectioned to 3 μm thickness using an ultramicrotome (EM UC7i, Leica) with a glass knife. Sections were submerged in 0.1% (w/v) toluidine blue O solution for 5 sec. For GUS-stained samples, sections were counterstained with the periodic acid-Schiff reaction, according to the following procedures: submerging into 1% (w/v) periodic acid for 4 min, washing with water, submerging into Schiff's reagent (Merk) for 10 sec, and washing with water. Photographs of sections were captured using a Scanscope CS2 scanner (Leica).

**Preparation of serial sections and 3D reconstruction.** Samples were embedded in Technovit 7100 resin (Heraeus Kulzer GmbH) according to the abovementioned method and were sectioned to 3 μm thickness using a microtome (RM2245, Leica) with a tungsten knife. Serial sections were stained with 0.1% (w/v) calcofluor white (Fluorescent Brightener 28) for 30 min, and then the fluorescence of calcofluor white was excited at a wavelength of 405 nm and detected at wavelengths from 415 to 475 nm by a confocal laser-scanning microscope (TCS SP8, Leica) with a Leica Hybrid Detector (HyD). Using Fiji/TrakEM2 software[38], images of serial sections were aligned, and the tissue was painted with false colors. A 3D image was generated using the ImageJ 3D viewer program (resampling = 2, smooth = 5 iterations).

**Modified pseudo-Schiff propidium iodide method.** Samples were fixed with ice-cold Farmer's solution (3:1 [v/v] ethanol: acetic acid) at 4 °C overnight and then rehydrated through a graded ethanol series of 70, 50, 30% (v/v) ethanol solution and distilled water for 10 min each. The rehydrated samples were submerged into mixtures composed of 0.2 M NaOH and 1% (w/v) sodium dodecyl sulfate at 37 °C overnight and then incubated in 0.01% (w/v) α-amylase solution in phosphate buffer at 37 °C overnight. The modified pseudo-Schiff propidium iodide method was performed as described in Truernit et al.[39]. Samples were soaked in 1% (w/v)

periodic acid for 40 min and then in a mixture of 100 mM sodium metabisulfite, 100 mg/ml propidium iodide, and 0.15 M HCl at room temperature for 90 min. Stained samples were embedded in a clearing solution (4:1:2 [w/v/v] chloral hydrate:glycerol:water) and observed using a confocal laser-scanning microscope (TCS SP8, Leica) with an HyD detector at an excitation wavelength of 488 nm and an emission wavelength range of 550–720 nm.

**Phloroglucinol staining**. Cross-sections were prepared freehand using a single-edge razor blade from the hypocotyls of bolted plants, fixed with ice-cold Farmer's solution (3:1 [v/v] ethanol: acetic acid), and rehydrated through the above-mentioned ethanol series. The rehydrated sections were stained with 1% (w/v) phloroglucinol in 6 M HCl for 10 min and then embedded in 50% glycerol in 3 M HCl. Photographs were obtained using a stereomicroscope (M165FC, Leica) with an MC120 HD camera (Leica).

**Maceration of xylem**. Plant samples were harvested from the upper end of the incision at 7 DAI and then incubated in a mixture (1:1 [v/v] glacial acetic acid:30% hydrogen peroxide) at 60 °C for 3 days[40]. After washing with distilled water, macerated samples were soaked in clearing solution (4:1:2 [w/v/v] chloral hydrate: glycerol:water) and gently pipetted up and down to separate the cells. Differential interference contrast images were captured using a confocal laser-scanning microscope (TCS SP8, Leica).

**EdU-labeling**. Flowering stems were incised as described above and then plant pots were transferred to a container (14.2 × 20.6 × 6.6 cm, width × length × height) filled with 2.5 μM EdU solution and grown for 5 days. Plant samples were harvested at 5 DAI, fixed with Farmer's solution and then washed three times in Phosphate-buffered saline. Coupling of EdU to the Alexa Fluor substrate was performed according to the manufacturer's instructions (Click-iT EdU Alexa Fluor 488 imaging kit, Thermo Fisher, USA). Before observations, samples were embedded with Super Cryoembedding Medium (Section-Lab, Japan), cryo-sectioned long-itudinally into 80–100 μm-thick using a cryostat (CM1860, Leica), and then stained by DAPI (NucBlue Fixed Cell Ready Probes Reagent, Thermo Fisher). The double-stained sections were observed under a confocal laser-scanning microscope (TCS SP8, Leica), nuclear stained by DAPI and Alexa 488–labeled EdU were excited at 405 or 488 nm, respectively.

**VISUAL assay in cotyledons**. Seeds were surface-sterilized and vernalized as described above, then sown and grown on rectangular plastic plates containing half-strength Murashige and Skoog medium[41] and 0.25% (w/v) sucrose in 1.5% (w/v) solidified agar under white light (60 μmol m$^{-2}$ s$^{-1}$) at 22 °C for 6 days. Induction assays of xylem cells were performed as described in Kondo et al.[20]. Hypocotyls were cut off at the middle, and then cotyledons were transferred to half-strength Murashige and Skoog[41] liquid medium containing 1.25 mgl$^{-1}$ 2,4-D and 0.25 mgl$^{-1}$ kinetin with or without 10 μM bikinin in 12 multiwell plates (Sumilon) and incubated under the same growth conditions with shaking at 120 rpm. Cotyledons were fixed, rehydrated, and embedded in a clearing solution (4:1:2 [w/v/v] chloral hydrate:glycerol:water). Cleared cotyledons were observed using a fluorescent optical microscope (BX53, Olympus) with a DP73 camera (Olympus), and UV images were obtained with a fluorescence filter cube (U-FUNA filter from Olympus; 360–370 nm excitation filter, 420–460 nm emission filter, 410 nm dichroic mirror).

**Phylogenetic analysis**. The amino acid sequence of ANAC071 (DDBJ BAW35238.1) was used to search for homologous NAC genes using BLAST programs. BLAST searches were performed in the following sites: *Arabidopsis thaliana*, (http://www.arabidopsis.org/Blast/); *Solanum lycopersicum* and *Nicotiana tabacum*, (https://solgenomics.net/tools/blast/); *Lotus japonicus*, (http://www.kazusa.or.jp/lotus/blast.html); *Vitis vinifera*, (http://www.genoscope.cns.fr/cgi-bin/blast_server/projet_ML/blast.pl)[42]; *Populus trichocarpa*, (http://popgenie.org/blast); *Oryza sativa*, *Hordeum vulgare*, and *Zea mays*, (https://plants.ensembl.org/Multi/Tools/Blast); *Phyllostachys heterocycle*, (http://202.127.18.221/bamboo/blast.php)[43]; *Amborella trichopoda*, (http://amborella.huck.psu.edu/); *Marchantia polymorpha*, (http://marchantia.info/blast/). Multiple sequence alignment of NAC genes was conducted using the online version of MAFFT with the L-INS-I option (https://mafft.cbrc.jp/alignment/server/)[44], and the combined sequences around the NAC domain (112 amino acids; Supplemental Dataset 1) were used to draw the phylogenetic tree with MEGA version 7.0 software (Neighbor-Joining method; 1000 bootstrap replications; JTT substitution model)[45].

**Statistics and reproducibility**. The statistics, the number of biological replications and plant samples are described in the method or figure legend of the corresponding section. Gene expression analyses were performed using at least three biologically independent samples. For the quantification of cross-section or microscope images, at least two biologically independent samples were used.

**Reporting summary**. Further information on research design is available in the Nature Research Reporting Summary linked to this article.

## Data availability
All data and genetic material used for this paper are available from the corresponding author upon request. All section images for Fig. 4b and all leaf images for Fig. 5b are available in Supplementary Data 1. The source data used to generate graphs in main figures are available in Supplementary Data 2.

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

## Acknowledgements

We are grateful to Prof. H. Yamane, Prof. T. Yokota, and Dr. K. Miyamoto (Teikyo University, Japan) for valuable suggestions and technical assistance. We also thank Dr. S. Naramoto (Hokkaido University, Japan) for valuable information about EdU labeling. This work was supported in part by the Japan Society for the Promotion of Science (Grant-in-Aid for Young Scientists B no. 16K18572, Grant-in-Aid for Scientific Research (C). no. 19K06728 to M.A.), the Ministry of Education, Culture, Sports, Science and Technology of Japan (Grants-in-Aid no. 17H06476 to Yu.K.), the Ministry of Education, Culture, Sports, Science, and Technology Program for the Strategic Research Foundation at Private Universities (grant no. S1311014 to M.A.), and the Promotion and Mutual Aid Corporation for Private School of Japan (to M.A.).

## Author contributions

K.M. conceived the original research plan. K.M. and M.A. wrote the article, designed the experiments. K.M., M.A., and S.S. supervised the overall study. Y.M. and Yo.K. performed the microscopic experiments in the analysis of mutants. R.S., H.I., N.N., and K.S. perfomed histochemical analysis using promoter GUS lines. K.M. and Yu.K. analyzed VISUAL assay. R.S., K.S., and M.A. performed EdU assay. R.S., Yu.K., and S.S. assisted with the writing.

## Competing interests

The authors declare no competing interests.
