## [Peer Review File · Communications Biology]

Reviewers' comments:

Reviewer #1 (Remarks to the Author):

In the manuscript by Matsuoka et al., the authors investigated the cambium formation during tissue reunion process after stem incision and they explored the roles of previously identified transcription factor ANAC071 and its homologous genes ANAC096 and 011 during this process. They utilised both in vivo stem incision and in vitro cell culture systems to study the functions of ANACs and they claim that both ANAC071 and 096 are required for cell divisions associated with wound-induced cambium formation during tissue-reunion process.

I have a few comments:

Major points:

- To investigate the gene expression levels in various conditions, the authors utilised both qRT-PCR and GUS reporters. The authors should take the advantages of GUS reporter lines (rather than the qRT-PCR) to demonstrate the temporal-spatial dynamics of ANACs and cambium markers (such as TDR) during the process of wound cambium formation and xylem re-differentiation. It would be more convinced if the authors could use the reporter lines to show that ANAC071 and 096 are induced at 1 DAI whereas TDR is induced later on. On a related note, cell proliferation associated with the "cambialization" process is central part of this m/s. However, there is very little direct evidence of cell proliferation (such as reporter analysis or EDU staining) during this process. The conclusions are only based on histological analysis, and it is thus unclear whether ANAC expression upregulation come up first, before cell proliferation (as expected based on the model).
- In current manuscript, there are mixed use of many terms: wound-induced cambium/cambium, intravascular wound-induced cambium and extravascular wound-induced cambium (I don't see any data for this), wound-induced secondary xylem and inter-vascular (Supp Fig. 12b, typo?) wound-induced cambium. The author should label them clearly in the figures which types of cambium they referred to. In particular, please separate the wound-induced cambium and xylem. For example, it is inappropriate to mark all tissues as Iwc (intravascular wound-induced cambium) in Fig. 1j. Also, it is important to mark intravascular and extravascular regions in Fig. 1 clearly. Finally, what is "differentiated xylem parenchyma" - shouldn't parenchyma be undifferentiated by its definition?
- Related to my last two comments, in the stem incision system, there should be two types of wound-induced cambium based on the authors' model: extravascular (originated from parenchyma cells of pith and protoxylem) and intravascular (originated from parenchyma cells in the meta- or secondary xylem that are adjacent to the existed procambium). However, the authors mostly reported the phenotypes for the later types, that is "Iwc" in all the genotypes. Did the authors analyze the extravascular cambium phenotypes and if so what are the phenotypes?
- First, the authors should label clearly the division of pith cells and protoxylem parenchyma cells on the sections (Fig. 1 and Supplementary Fig. 5a-f). Second, to me, there is still cell divisions in the double and triple mutants compared to the decapitated samples. The authors have developed the 3D-imaging approach to identify the cell types of dividing cells upon incision, such technique could be used to characterize the mutants' phenotypes. Additionally, the authors should quantify the cell divisions to reach the conclusion.

Minor points:

- TDR was first called PXY, so it is important to mention that name alongside TDR, and cite the appropriate paper(s) in the introduction.
- In Fig 2, the authors stated "Sampling time at 0 DAI means stem before incision", please describe the position of the 0DAI sample, is it as the same height of the incised region (black box) or 50mm above the incised region (white box)? It is possible that the expression levels of these ANAC genes are spatially different, for example, they could show higher expression in lower part of the stems. Another way to prove this is to demonstrate their expression patterns along the intact stem using the reporter

lines shown in Fig.3.

- In discussion authors indicate the central role of xylem initial cells between cambium and differentiated vessels in the "cambialization" process. Recently, Smetana et al (2019) suggested that these xylem initial cells acts as cambium stem cell organiser. Given that these cells were defined by local auxin maximum and subsequent TDR/PXY expression, this would make an interesting link to the "cambialization" process as presented in this study? Does this process require dedifferentiation, or is it just activation of the stem cell organizer to overproliferate?
- No GUS data for the expression of ANAC071 and ANAC096 in stomata or cut hypocotyls as the author claimed.
- Please, show better images of cross sections for ANAC096::GUS (Fig. 3k) to reveal its expression in parenchyma cells.
- Line 108, when the authors stated "cortex cells began elongation...", did they mean expansion?

Reviewer #2 (Remarks to the Author):

This paper demonstrated, with detail, that the function of the transcription factors ANAC071 and ANAC096 is related to the induction of cambium formation after incision using the double mutant. Results also include a triple mutant, *anac071 096 011*, and demonstrated that ANAC011 is not critical for tissue reunion meanwhile ANAC071 and ANAC076 functions are redundant.

In general, the manuscript is well written, figures are nice with a clear description and discussion. Almost all the studies were done either in the double and triple mutants although there were no significant differences between them. So it is confusing that fig 7 only includes results from triple mutant.

Fig7 could include the results of the double mutant.

Supp Fig 12 should be included in the main text.

Incise c is not described in the figure legend of supp fig 12.

Responses for reviewers' comments:

manuscript#: COMMSBIO-20-0091-T

Title: Wound-inducible ANAC071 and ANAC096 transcription factors promote cambial cell formation in incised Arabidopsis flowering stems.

We greatly appreciate reviewers for careful reading and helpful comments. Those comments helped us improve our manuscript and provided important guidance for our research.

The revised points we have made are as follows:

Reviewer 1

- 1. To investigate the gene expression levels in various conditions, the authors utilised both qRT-PCR and GUS reporters. The authors should take the advantages of GUS reporter lines (rather than the qRT-PCR) to demonstrate the temporal-spatial dynamics of ANACs and cambium markers (such as TDR) during the process of wound cambium formation and xylem re-differentiation. It would be more convinced if the authors could use the reporter lines to show that ANAC071 and 096 are induced at 1 DAI whereas TDR is induced later on. On a related note, cell proliferation associated with the “cambialization” process is central part of this m/s. However, there is very little direct evidence of cell proliferation (such as reporter analysis or EDU staining) during this process. The conclusions are only based on histological analysis, and it is thus unclear whether ANAC expression upregulation come up first, before cell proliferation (as expected based on the model).**

Thank you for taking time and meaningful suggestions. According to reviewer's comments, we have included histochemical analysis of ANAC071, 096 and TDR/PXY using GUS reporter lines (Supplementary Fig. 2), DAPI-staining and EdU-labeling experiments (Supplementary Fig. 7) and added a brief explanation and information about these experiments in revised manuscript.

- 2. In current manuscript, there are mixed use of many terms: wound- induced cambia/cambium, intravascular wound-induced cambium and extravascular wound-induced cambium (I don't see any data for this), wound-induced**

secondary xylem and inter-vascular (SuppFig. 12b, typo?) wound-induced cambium. The author should label them clearly in the figures which types of cambium they referred to. In particular, please separate the wound-induced cambium and xylem. For example, it is inappropriate to mark all tissues as Iwc (intravascular wound-induced cambium) in Fig. 1j. Also, it is important to mark intravascular and extravascular regions in Fig. 1 clearly.

Thank you for pointing out. We agree that the characterization of wound-induced cambium is an important line of study. however, at this moment, we don't have enough data to compare the wound-induced cambium and wound-induced secondary xylem, and intravascular wound-induced cambium and extravascular wound-induced cambium at cellular and molecular levels. We want to investigate these in next studies.

So, we have changed “wound induced extravascular cambium” to “wound induced cambium of extravascular region”, “wound induced intravascular cambium” to “wound induced cambium of intravascular region” and “Iwc (intravascular wound-induced cambium)” to “Wic (wound-induced cambium).” (P.9 line 126-127, P.9 line 164-166; Fig.1, Fig.3, Fig.7, Fig.8)”, and revised them in the entire manuscript.

- 3. Finally, what is “differentiated xylem parenchyma” - shouldn't parenchyma be undifferentiated by its definition?**

We removed “differentiated parenchyma cells” (P.25 line 377).

- 4. Related to my last two comments, in the stem incision system, there should be two types of wound-induced cambia based on the authors' model: extravascular (originated from parenchyma cells of pith and protoxylem) and intravascular (originated from parenchyma cells in the meta- or secondary xylem that are adjacent to the existed procambium). However, the authors mostly reported the phenotypes for the later types, that is “Iwc” in all the genotypes. Did the authors analyze the extravascular cambium phenotypes and if so what are the phenotypes?**

Thank you for suggestion. we don't have enough cellular and molecular data for extravascular cambium phenotypes at this moment. It would be interesting to show

detailed phenotypes the wound induced extravascular cambium as well as intravascular cambium. The data that we have cannot reveal these, we would like to carry out these analyses in our continuing research. We consider that experiments using reporter analysis, multiple mutants and complementation test may also be necessary in order to fully understand the phenotypes for the extravascular/intravascular cambium.

In order to better explain that cambium-like cells were found both in intravascular region and extravascular region, we changed “ In this study, we refer to these cambiums as the intravascular wound-induced cambium and extravascular wound-induced cambium”. to “These TDR/PXY-expressing cambium-like cells were found both in intravascular region and extravascular region.” (P.9 line 126-127).

To make clearly understandable, we also changed “ wound induced extravascular cambium” to “ wound induced cambium of extravascular region” and “ wound induced intravascular cambium” to “ wound induced cambium of intravascular region” in revised manuscript.

5. First, the authors should label clearly the division of pith cells and protoxylem parenchyma cells on the sections (Fig. 1 and Supplementary Fig. 5a-f).

Thank you for pointing out. We added symbol for cortex, vascular bundle and pith in Supplementary Fig. 6a-f, but it is difficult to distinguish between dividing pith cells and protoxylem parenchyma cells from section. So, we would like to remain Fig. 1 as it is.

6. Second, to me, there is still cell divisions in the double and triple mutants compared to the decapitated samples. The authors have developed the 3D-imaging approach to identify the cell types of dividing cells upon incision, such technique could be used to characterize the mutants' phenotypes. Additionally, the authors should quantify the cell divisions to reach the conclusion.

Thank you for suggestion of meaningful experiments, but we have not been able to include these experiment in this manuscript. The 3D developing method experiment is very interesting and meaningful experiments to fully understand the mutant's

phenotypes, but in this paper we prefer to focus on the cambial cell proliferation and vascular cell formation, so we would like to carry out these analyses in our continuing research.

In order to quantitatively compare cell proliferation of WT and *anac* mutants, the number of 7DAI cell layers was mentioned in Fig.4, but we have now also performed DAPI-staining and EdU-labeling experiments (Supplementary Fig. 7), and added the following sentence for clarity, “Hence, the cell layers of these wound-induced cambial cells in the upper region of the incision were also quantitatively compare (Fig. 4), and DAPI-staining and EdU-labeling experiments were also performed to confirm the cell proliferation (Supplementary Fig. 7).” (P.13, line 194-197) and “In the incised region of WT at 5DAI, higher number of DAPI-stained nuclei were observed, and these cells were strongly labeled with EdU (Supplementary Fig. 7). On the other hand, relatively small numbers of DAPI-stained nuclei with EdU-labeling were detected in incised stem of *anac071 096 011* triple mutants and non-incised region of WT (Supplementary Fig. 7).” (P.14, line 204-208).

Minor points:

- 7. TDR was first called PXY, so it is important to mention that name alongside TDR, and cite the appropriate paper(s) in the introduction.**

Thank you. We revised them in the manuscript.

- 8. In Fig 2, the authors stated “Sampling time at 0 DAI means stem before incision”, please describe the position of the 0DAI sample, is it as the same height of the incised region (black box) or 50mm above the incised region (white box)? It is possible that the expression levels of these ANAC genes are spatially different, for example, they could show higher expression in lower part of the stems. Another way to prove this is to demonstrate their expression patterns along the intact stem using the reporter lines shown in Fig.3.**

Thank you for suggestion. We added information about experimental procedure as “harvested from same height as incised region; black box in a”(P.45 line 694) and

included the results of GUS reporter lines (Supplementary Fig. 2).

- 9. In discussion authors indicate the central role of xylem initial cells between cambium and differentiated vessels in the “cambialization” process. Recently, Smetana et al (2019) suggested that these xylem initial cells acts as cambium stem cell organiser. Given that these cells were defined by local auxin maximum and subsequent TDR/PXY expression, this would make an interesting link to the “cambialization” process as presented in this study? Does this process require dedifferentiation, or is it just activation of the stem cell organizer to overproliferate?**

Thank you for suggestion. We cited the paper Smetana et al. (2019) and Hajný et al. (2020) and added brief explanation in text.

We corrected explanation about cambialization as “... which means the conversion of differentiated cells to cambial cells and /or activation of cambial cell proliferation in the tissue repair process of incised flowering stems” (P.20 line 302-303).

We also changed sentences as follows.

“ANAC071 and ANAC096 are essential for cambialization during the tissue-reunion process” to “ANAC071 and ANAC096 are essential for cell proliferation during the tissue-reunion process.” (P.13 line 306).

“ANAC071 and ANAC096 are involved in the cambialization process in the VISUAL assay” to “ANAC071 and ANAC096 are involved in the ectopic vascular cell differentiation.” (P.23 line 348).

“... the process of cambialization from differentiated parenchyma cells; however, the molecular mechanisms underlying the regulation of ectopic differentiation into cambial cells and of vascular differentiation should be further examined.” to “...process of cambialization in incised flowering stem; however, direct evidence to support cell dedifferentiation/ redifferentiation, and the molecular mechanisms underlying the regulation of ectopic differentiation into cambial cells, cell proliferation and vascular differentiation, should be further examined.” (P.13 line 376-379).

- 10. No GUS data for the expression of ANAC071 and ANAC096 in stomata or cut hypocotyls as the author claimed.**

We corrected the text.

- 11. Please, show better images of cross sections for ANAC096::GUS (Fig. 3k) to reveal its expression in parenchyma cells.**

We improved the resolution of images (Fig. 3k, i).

- 12. Line 108, when the authors stated “cortex cells began elongation...”, did they mean expansion?**

We changed “cortex cells began elongation..” to “cortex cells began expansion” (P.8 line 114).

Reviewer #2 (Remarks to the Author):

This paper demonstrated, with detail, that the function of the transcription factors ANAC071 and ANAC096 is related to the induction of cambium formation after incision using the double mutant. Results also include a triple mutant, anac071 096 011, and demonstrated that ANAC011 is not critical for tissue reunion meanwhile ANAC071 and ANAC076 functions are redundant. In general, the manuscript is well written, figures are nice with a clear description and discussion.

We appreciate taking time and offering the helpful suggestions.

- 1. Almost all the studies were done either in the double and triple mutants although there were no significant differences between them. So it is confusing that fig 7 only includes results from triple mutant. Fig7 could include the results of the double mutant.**

Thank you for comments. As shown in Fig.4 and Supplementary Fig. 5, there was no significant difference between the anac071 096 double mutant and the anac071 096 011 triple mutant, but ANAC011 was located in same clades of ANAC071 and

ANAC096 (Supplementary Fig.3). So, we generated pTDR::GUS reporter line using both wild type and anac071 096 011 triple mutant background and performed corresponding studies.

2. Supp Fig 12 should be included in the main text. Incise c is not described in the figure legend of supp fig 12.

Thank you for suggestions. We moved Supplementary Fig .12 to main text as Fig. 8, and corrected figure legend.

“Additional correction”

1. We corrected the sentence as follows; “(a-c) ...Welch's test (*P < 0.05. ** P < 0.05).” to “(a-c) ...Welch's test (*P < 0.05. ** P < 0.01).” (Fig. 6; P.48 line 731).
2. We added the sentence “Relative gene expression was calculated when WT 0h was set to 1.” (Fig. 6; P.48 line 734-735 and Supplementary Fig.10).
3. We have added Dr. Ryosuke Sato and Ms. Kyomi Shibata who performed histochemical analysis and EdU assay as co-author and corrected acknowledgements, author contributions, reference and figure numbers.

We again appreciate the helpful suggestions offered by the reviewers, as their comments were very useful for revising this manuscript. Thank you for giving us the opportunity to strengthen our manuscript with your valuable comments and suggestions.

Sincerely yours,

Masashi Asahina
Department of Biosciences,
Teikyo University.
1-1 Toyosatodai, Utsunomiya,
Tochigi, 3208551, JAPAN.
asahina@nasu.bio.teikyo-u.ac.jp

REVIEWERS' COMMENTS:

Reviewer #1 (Remarks to the Author):

The authors have raised all of my concerns. Thus, I have no further comments.

Reviewer #2 (Remarks to the Author):

After reviewing the corrections made to the manuscript "Wound-inducible ANAC071 and ANAC096 transcription factors promote cambial cell formation in incised Arabidopsis flowering stems." by Dr Asahina and colleagues, I consider that they have answered all the doubts and comments that had been made.